# Gene Expression Profile of Mexican Lime (*Citrus aurantifolia*) Trees in Response to Huanglongbing Disease caused by *Candidatus* Liberibacter asiaticus

**DOI:** 10.3390/microorganisms8040528

**Published:** 2020-04-07

**Authors:** Ángela Paulina Arce-Leal, Rocío Bautista, Edgar Antonio Rodríguez-Negrete, Miguel Ángel Manzanilla-Ramírez, José Joaquín Velázquez-Monreal, María Elena Santos-Cervantes, Jesús Méndez-Lozano, Carmen R. Beuzón, Eduardo R. Bejarano, Araceli G. Castillo, M. Gonzalo Claros, Norma Elena Leyva-López

**Affiliations:** 1Instituto Politécnico Nacional, CIIDIR-Unidad Sinaloa, 81101 Guasave, Mexico; angela_paulina22@hotmail.com (Á.P.A.-L.); msantos@ipn.mx (M.E.S.-C.); jmendezl@ipn.mx (J.M.-L.); 2Plataforma Andaluza de Bioinformática, Universidad de Málaga, 29590 Malaga, Spain; rociobm@uma.es (R.B.); claros@uma.es (M.G.C.); 3CONACyT, Departamento de Biotecnología Agrícola, Instituto Politécnico Nacional, CIIDIR-Unidad Sinaloa, 81101 Guasave, Mexico; edgarrnegrete@gmail.com; 4Campo Experimental Tecomán-INIFAP, Carretera Colima-Manzanillo km. 35. Tecomán, 28100 Colima, Mexico; 5Área de Genética, Facultad de Ciencias, Instituto de Hortofruticultura Subtropical y Mediterránea La Mayora (IHSM-UMA-CSIC), Universidad de Málaga, 29010 Málaga, Spain; 6Departamento de Biología Molecular y Bioquímica, Universidad de Málaga, 29010 Malaga, Spain

**Keywords:** HLB, *Candidatus* Liberibacter asiaticus (CLas), transcriptomics, tolerant citrus species, Mexican lime

## Abstract

Nowadays, Huanglongbing (HLB) disease, associated with *Candidatus* Liberibacter asiaticus (CLas), seriously affects citriculture worldwide, and no cure is currently available. Transcriptomic analysis of host–pathogen interaction is the first step to understand the molecular landscape of a disease. Previous works have reported the transcriptome profiling in response to HLB in different susceptible citrus species; however, similar studies in tolerant citrus species, including Mexican lime, are limited. In this work, we have obtained an RNA-seq-based differential expression profile of Mexican lime plants challenged against CLas infection, at both asymptomatic and symptomatic stages. Typical HLB-responsive differentially expressed genes (DEGs) are involved in photosynthesis, secondary metabolism, and phytohormone homeostasis. Enrichment of DEGs associated with biotic response showed that genes related to cell wall, secondary metabolism, transcription factors, signaling, and redox reactions could play a role in the tolerance of Mexican lime against CLas infection. Interestingly, despite some concordance observed between transcriptional responses of different tolerant citrus species, a subset of DEGs appeared to be species-specific. Our data highlights the importance of studying the host response during HLB disease using as model tolerant citrus species, in order to design new and opportune diagnostic and management methods.

## 1. Introduction

Nowadays, Mexican lime (*Citrus aurantifolia*), belonging to *Rutaceae* family, is one of the most economically important citrus crops in several countries. Lime is thought to be native to the tropical areas in Southeastern Asia, even though its genetic origin is not clear and the (F1) interspecific hybrid nature of Mexican lime (*Citrus micrantha* and *Citrus medica*) [1] has been recently reported. In Mexico, the lime industry is considered one of the main activities of the primary sector in the Pacific region, both in terms of production volume and economic value. According to data from the Agri-Food and Fisheries Information Service (SIAP-SAGARPA) during 2018, the production of Mexican lime in Mexico on 95,177 ha was over one million tons, mainly distributed in the states of Michoacán, Colima, Oaxaca, and Guerrero [2], generating a revenue of more than 267 million USD. This industry creates jobs all year round, both in the management and harvesting of the crop and the processing of the fruit. Recently, the Mexican lime production has been impacted by different biotic factors such as viruses, fungi, and bacteria, causing diseases, reducing yield and fruit quality and, in many cases, causing plant/tree death [3].

Huanglongbing (HLB), also known as citrus greening, is considered one of the most devastating worldwide citrus diseases. HLB etiology is caused by the Gram-negative α-proteobacteria *Candidatus* Liberibacter spp. Taxonomically, there are three HLB-associated species. namely *Candidatus* Liberibacter asiaticus (CLas), *Ca*. L. africanus, and *Ca*. L. americanus, categorized according to their presumptive geographical origin (Asia, Africa, and America continents, respectively) and 16S rDNA molecular classification. CLas-associated HLB, vectored by the Asian citrus psyllid (*Diaphorina citri*) is the most prevalent worldwide, and in Mexico, additionally to CLas, *Ca*. Phytoplasma asteris has been associated at early stages of HLB disease development [4,5,6]. Early symptoms of HLB include yellowing of leaves followed by dieback of both the canopy and fibrous roots. HLB leads to premature plant decline and excessive yield loss, which results in heavy economical loss. Symptoms associated with HLB disease are considered the result of deregulation of physiological, cellular, and molecular processes and an exacerbated host antibacterial response [7].

Identifying potential molecular targets of CLas has been particularly challenging, as there is no method for maintaining CLas in culture in laboratory conditions; therefore, the knowledge pertaining to its physiology has been derived from in silico predictions based on the information encoded in its genome [8]. Conventionally, HLB disease management resides in chemical control to reduce the psyllid vector population and agronomical practices, including removal of infected trees, production of pathogen-free plants, and nutritional management [9]. Therefore, the ultimate long-term strategy for HLB control relies on the developing of new cultivars with HLB resistance. Unfortunately, to date, no resistance genes have been described in citrus species, making traditional breeding challenging [4]. Alternatively, transgenesis and CRISPR-mediated genome editing technologies have shown promising potential in generating disease-resistant citrus varieties [10]. However, both transgenic and genome editing requires basic knowledge about host defensive responses to bacterial infection as well as about the pathogenic mechanism of the invading pathogen. All known citrus species can be affected by HLB disease; however, they show significantly different host responses to infection and symptom development in greenhouse and field conditions. Generally, sweet orange and grapefruit are susceptible, and some lemons and limes including Mexican lime have been described as tolerant to HLB [11,12,13]; therefore, it is imperative to characterize the molecular responses to CLas infection in order to identify putative tolerance-related genes.

Transcriptomic profiling represents the first step to elucidate the molecular mechanisms of host–pathogen interactions, and consequently, the identification of target genes useful for the design of new diagnostic and management strategies. In order to elucidate the HLB tolerance showed by some citrus species, several transcriptomic studies of citrus–CLas interaction have been performed through microarray and high-throughput technologies. Transcriptomic profiles of different citrus tissues, including leaves, fruits, and roots from susceptible citrus cultivars challenged against CLas infection, showed that genes involved in sugar and starch metabolism, photosynthesis, cell wall metabolism, stress responses, and hormone signaling were significantly deregulated [9,14,15,16,17,18]. On the other hand, comparative transcriptomic studies between HLB tolerant and susceptible citrus species were performed to identify genes involved in disease tolerance. The tolerance in breeding line US-897 (*C. reticulata* x *P. trifoliata*) is associated with constitutive higher expression of defense-related genes compared with susceptible “Cleopatra” mandarin (*C. reticulata*) [19]. In response to CLas infection, a comparative transcriptomic analysis between tolerant (rough lemon, *C. jambhiri*) and susceptible (sweet orange, *C. sinensis*) lines showed that at early stages of the disease, more differential expressed genes are present in HLB-infected rough lemon than in sweet orange, but significantly fewer at late stages of the infection. Genes coding for cell wall proteins were identified as putative key factors in tolerance response [17]. In another study, the differential transcriptomic response to CLas infection between tolerant “Jackson” grapefruit and susceptible “Marsh” grapefruit identified genes involved in secondary metabolism, pathogenesis, transcription factors, hormone signaling, and receptor-like kinases as potential factors involved in tolerance [20]. Recently, in a study comparing the profiles of gene expression of tolerant Kaffir lime (*C. hystrix*) and susceptible *C. sinensis*, in addition to cell wall and secondary metabolism genes, activation of transcripts coding peroxidases were associated to tolerance [21]. Together, these reports suggest that despite some cellular responses being common between tolerant citrus cultivars, other responses are species-specific, highlighting the importance of transcriptomic studies in response to CLas infection in tolerant citrus species. In order to understand how CLas interacts with Mexican lime at a molecular level, a greenhouse rigorous time-course experiment avoiding environmental factors and individual differences among plants was designed. In this study we report the transcriptomic profile of leaves from the acid lime *C. aurantifolia* (Mexican lime) challenged against CLas infection at both asymptomatic and symptomatic stages of HLB disease severity.

## 2. Materials and Methods

### 2.1. Plant Material and Experimental Design

Mexican lime (*C. aurantifolia*) plants on alemow (*C. macrophylla*) rootstock used in this study were kept in a pathogen-free shadow-greenhouse at Experimental Station Tecoman-INIFAP, Tecoman, Colima, Mexico, with an average annual temperature of 26 °C (range 16–34 °C) and a mean humidity of 64% (range 46%–100%). A flowchart used for CLas inoculation and selection of samples for RNA-seq analysis is summarized in Figure 1. Budwoods from HLB-infected Mexican lime trees were used to inoculate a total of 45 Mexican lime plants (nine months old), and 15 other plants were inoculated with budwood from healthy Mexican lime plants as controls (mock-inoculated) (Figure 1a). After inoculation, plants were maintained at shadow-greenhouse conditions and fertilized when necessary. Eight complete leaves were collected from each plant at 8 and 16 weeks post-inoculation (wpi). In all cases, the fourth leaf of 8 different branches located at the third level of ramification from the main stem was collected in order to obtain a homogeneous representation of source tissue (Figure 1b). As tissue source for RNA-Seq analysis, for each sampled plant, 4 complete leaves were frozen with liquid nitrogen and ground with mortar and pestle. For bacteria quantification (qPCR), central midribs and petioles (tissue were bacteria is mainly located) were dissected from the remaining 4 collected leaves (Figure 1b). A total of 8 libraries were generated. For each library, 5 individual plants were selected according to homogeneity on the disease symptoms and bacterial titers. We generated libraries L8wpiHLB+1 and L8wpiHLB+2, each one containing 5 pooled 8 wpi asymptomatic plants, and libraries L16wpiHLB+5 and L16wpiHLB+5, each one containing 5 pooled 16 wpi symptomatic plants. Control libraries, L8wpiHLB-3 and L8wpiHLB-4 (for 8 wpi stage) and L16wpiHLB-7 and L16wpiHLB-8 (for 16 wpi stage), were constructed by pooling 5 mock-inoculated plants, respectively (Figure 1c). 

### 2.2. Determination of CLas Titer by Quantitative PCR (qCPR)

CLas titer in grafting inoculated *C. aurantifolia* plants was obtained by quantitative PCR (qPCR) using an absolute quantification protocol. About 200 mg of powdered tissue from midribs and petioles from each individual plants was used for total DNA extraction by a previously described CTAB protocol [22,23]. The qPCR reactions were performed in a 96-well plate using a CFX96TM Real-Time PCR Detection System (Bio-Rad) and EvaGreen^®^ Supermix according to the manufacturer’s protocol. The primer set HLBas/HLBr was used for PCR amplification to target the CLas 16S rDNA gene; furthermore, a primer set targeting a genomic *locus* of *COX* gene coding region, COX-F/COX-R was used as a normalizer [24] (Appendix A). The standard amplification protocol was 30 s at 95 °C and 40 cycles of 5 s at 95 °C, followed 10 s at 58 °C, and the melting curve program was 60 to 95 °C with a heating rate of 0.1 °C/s and a continuous fluorescence measurement. To create the standard curve for absolute quantification, serial dilutions of a plasmidic construct containing CLas 16S rDNA-derived amplicon, in the range of 10^2^–10^6^ molecules/μL, were obtained. Quantification cycle (Cq) values of each of the three technical replicates and the corresponding amount of DNA (Log10) were plotted, and standard curves were obtained by linear regression analysis. qPCR efficiency (E) was obtained by the formula E = 10^−1/*s*^, where a slope (s) of −3.32 represents an efficiency of 100%. Bacterial DNA quantification was obtained by Cq data extrapolation with standard curve equation. Cq values of COX gene internal control were used to normalize the data, and results were reported as number of bacterial cells/100 ng of total DNA. Bacterial titer quantification of the 45 CLas-infected plants was carried out at both 8 and 16 wpi stages. 

### 2.3. RNA Isolation and Sequencing

Total RNA was isolated from liquid nitrogen ground tissue of complete leaves of the selected individual plants according to TRIzol^®^ protocol (Thermo Fisher Scientific, Carlsbad, USA). The RNA yield and quality was spectrophotometrically verified assessing the A_280_/A_260_ ratio using a Nanodrop 2000 apparatus, and RNA integrity was evaluated by capillary electrophoresis using a 2100 Bioanalyzer RNA Nanochip (Agilent Technologies, Inc., Santa Clara, CA, USA). For each condition, total RNA from five selected plants was pooled in equimolar ratio to construct each cDNA library. The cDNA libraries containing ≈500 base pairs (bp) fragments were constructed using the TruSeq Stranded mRNA Sample Preparation kit (Illumina, San Diego, CA, USA) following the manufacturer´s instructions and sequenced separately (2 × 150 bp) on an Illumina NexSeq500 instrument MID-Output by Langebio-CINVESTAV, Irapuato facilities (Guanajuato, Mexico).

### 2.4. Differential Expression Analysis

RNA-Seq analysis was performed on Picasso, a supercomputer using OpenSUSE LinuxLEAP 42.3 with Slurm queue system and Infiniband FDR/QDR network (54/40 Gbps) consisting of 216 nodes with Intel E5-2670 2.6 GHz cores for a total of 3~456 cores and 22 TB of RAM. FastQC tools were applied to asses quality of raw reads using default parameters, and then contaminant sequences and low quality reads were removed using SeqTrimNext [25]. The leaves transcriptome of *Citrus aurantifolia* [26] was used as reference (it includes the Arabidopsis orthologs for sequences) for the mapping using Bowtie2 [27]. Transcript counts were obtained using Sam2count.py. Data normalization and differential expression studies were carried out using DEgenes-Hunter [28] with EdgeR and DESeq2 package; any differentially expressed candidate gene must appear in both algorithms with |log2ratio| ≥ 1 and false discovery rate (FDR) <0.05. Enriched gene ontology term annotations were identified using Singular Enrichment Analysis (SEA) in agriGO v.2 [29]. SEA calculates the gene ontology (GO) term enrichment by comparing the frequency of association of members of a set of genes with different GO terms relative to a reference set of genes. As a reference, precalculated set “Arabidopsis genome locus (TAIR10)” in agriGO was used. For comparison to Plant GO slim, we limited terms under “Advanced options”. The remaining parameters were default set (using Fisher and Yekutieli as statistical and multi-test adjustment methods, respectively, and 5 as the minimum number of mapping entries). The enrichment analysis was supplemented by REVIGO (available online: http://revigo.irb.hr/) visualization toolbox. Functions of differentially expressed genes into specific pathways were visualized using MapMan Software [30].

### 2.5. Gene Expression Validation

To evaluate the reliability of DEGs screened by RNA-Seq, 10 candidate deregulated genes (3 at early stage, 5 at late stage, and 2 at both stages) were validated by quantitative reverse transcription PCR (RT-qPCR). Based on the 10 DEGs sequences from cDNA libraries, specific primers were designed using Primer Select software (DNASTAR Lasergene, Madison, WI, USA). *COX* gene was used as internal control. Primers sequences used for gene expression validation are summarized in (Appendix A). For RT-qPCR, individual plants from the 8 RNA-Seq libraries (5 plants for library) were analyzed. For RNA-Seq, each condition was represented by 2 biological replicates; therefore, for RT-qPCR, 10 plants for each condition were analyzed. One microgram of total RNA from each individual plant was pre-treated with DNAse I (Promega, Madison, WI, USA) and then used for cDNA synthesis with reverse transcription kit Super-Script III Reverse Transcriptase (Thermo Fisher Scientific, Carlsbad, CA, USA). Quantitative PCR (qPCR) analysis was performed using a CFX96TM real-time PCR system (Bio-Rad) and SsoFast EvaGreen^®^ Supermix (Bio-Rad, Foster City, CA, USA) according to the manufacturer’s protocol. Relative expression levels of selected target genes were calculated by means of the 2^−ΔΔCt^ method [31] using the Cq value of COX gene as internal reference control. For each sample, the fold change value was obtained from the ratio of relative expression data between HLB-infected versus the corresponding mock-inoculated control, and the data converted to Log2 base. All qPCR reactions were analyzed using 3 technical replicates, and results were plotted as the average of fold change expression data from the 2 biological replicates analyzed for each condition (10 plants), and compared with the corresponding data in Log2 base from RNA-Seq.

## 3. Results

### 3.1. Selection of Samples for RNA-seq Analysis

For selection of samples for RNA-seq analysis, HLB disease progress was recorded and bacterial titer quantification of the individual 45 CLas-infected plants was carried out at both 8 and 16 wpi stage. Central leaf midribs and petioles were used for DNA extraction since the bacteria is concentrated at phloematic tissues, increasing the detection sensitivity. After 8 wpi, 75% (33/45) of CLas-inoculated plants remained asymptomatic and phenotypically similar to control mock-inoculated plants, whereas the remaining 25% (12/45) showed early disease symptoms and were not used for RNA-Seq analysis. On the other hand, at 16 wpi, 100% (45/45) CLas-inoculated plants showed the typical disease symptoms, including leaf yellowing and asymmetric blotchy mottling (data not shown). For the early (asymptomatic) disease development stage (8 wpi), a total of 10 CLas-infected asymptomatic plants containing homogeneous bacterial titer (~2.5 × 10^2^ bacterial cells/100 ng of total DNA) were selected. For the late (symptomatic) disease development stage (16 wpi), the same 10 CLas-infected plants (symptomatic at this point) containing homogeneous bacterial titer (~1.4 × 10^4^ bacterial cells/100 ng of total DNA at 16 wpi) were used. No bacterium was detected in non-inoculated plants (Figure 2a). As control, 10 mock-inoculated randomly chosen plants were selected. A representative picture of leaves selected as source for RNA-Seq analysis is shown in Figure 2b. Total RNA from full leaves of five selected plants for each stage (CLas-infected, and healthy mock-inoculated plants, at early 8 wpi and late 16 wpi) was pooled in equimolar concentration for the construction of eight libraries (see Materials and Methods). 

### 3.2. Identification of Differentially Expressed Genes

A total of eight libraries were constructed and sequenced, two libraries representing biological replicates for each condition. Libraries L8wpiHLB+1 and L8wpiHLB+2 for 8 wpi at early (asymptomatic) stage, libraries L16wpiHLB+5 and L16wpiHLB+5 for 16 wpi at late (symptomatic) stage, and control libraries L8wpiHLB-3 and L8wpiHLB-4 (for 8 wpi stage) and L16wpiHLB-7 L16wpiHLB-8 (for 16 wpi stage). The raw data were deposited in the NCBI database (Accession number: SRR10353555, SRR10353556, SRR10353557, SRR10353558, SRR10353559, SRR10353560, SRR10353561, and SRR10353562). After filtering and trimming of low-quality and adapter sequences, we reached a sequencing depth of 21–29 million 300 bp paired-end reliable reads per library. The clean reads were aligned to our recently reported de novo assembled reference transcriptome of *Citrus aurantifolia* [26]. Differentially expressed genes (DEGs) were selected according to the analysis performed with EdgeR and DESeq2 package with |log2ratio| ≥ 1 and FDR < 0.05 as cut-off between mock-inoculated control plants and CLas-inoculated in asymptomatic (8 wpi) and symptomatic (16 wpi) stages. A graphical representation of the general gene expression (MA plot) is shown in Figure 3a and b (fold change in logarithmic scale versus global average of the normalized count of reads). A total of 2495 CLas-dependent differentially expressed genes (DEGs) were deregulated during asymptomatic stage, whereas in symptomatic stage, 4559 genes were significantly affected (Figure 3c). The complete list of fold change values of identified DEGs and their respective adjusted *p*-values are provided in Appendix A. Principal component analysis using DESeq2 package showed a distinct clustering of HLB-infected and control treatments with 86% of variance explained by PC1 and 10% of variance explained by PC2 for asymptomatic stage, whereas 92% of variance by PC1 and 5% of variance for PC2 was observed for symptomatic stage samples (Figure 4a,c). Similarly, a principal component analysis using EdgeR package showed that for both asymptomatic and symptomatic stages, biological replicates are located in the same domain (Figure 4b,d). These results indicate that a correlation exists between biological replicates demonstrating the reliability and consistency of transcriptional changes in the different conditions.

### 3.3. RNA-seq Data Validation by RT-qPCR

To validate the data obtained from RNA-Seq analysis, transcript expression of 10 selected genes was evaluated by quantitative reverse transcription PCR (RT-qPCR). The chosen genes (three deregulated at early stage, five at late stage and two at both stages) were selected from the significantly enriched pathways according PageMan analysis, including cell wall metabolism, hormone signaling, light reactions, secondary metabolism, biodegradation of xenobiotics, ubiquitin-dependent protein degradation, signaling, and redox reactions. The genes *CESA4* (Cellulose synthase A catalytic subunit 4) and *CYP79A2* (Cytochrome p450 79a2) belonging to cell wall metabolism pathways and biodegradation of xenobiotics family, respectively, were analyzed at both stages. At asymptomatic (early) stage, the genes *GES* (E,E-geranyllinalool synthase), *ABI1* (ABA Insensitive 1), and *PLC6* (Phosphoinositide phospholipase C 6), belonging to secondary metabolism of isoprenoids, hormone signaling pathways, and signaling of phosphinosites, respectively, were analyzed. Finally, at symptomatic (late) stage, the genes *LHCB5* (Chlorophyll a-b binding protein), *LAC7* (Laccase 7), *Thioredoxin-like 4* protein, *GSTU6* (Glutathione S-transferase TAU 6), and *EBF1* (EIN3-binding F-box protein 1), belonging to light reactions, secondary metabolism of phenols, redox reactions, and ubiquitin-dependent protein degradation, respectively, were analyzed. RT-qPCR analysis revealed expression patterns similar to the results obtained from RNA-seq for the 10 analyzed genes (Figure 5). To determinate the statistical significance of data obtained from RT-qPCR and RNA-seq, a chi-square test was performed to compare the fold-change results obtained from both methods. A *p*-value of 4.33 × 10^−7^ for the population of genes evaluated at both asymptomatic and symptomatic stages was observed, showing high correlation between data obtained from RT-qPCR and RNA-seq. These results confirm the accuracy and reliability of data generated from RNA-Seq analysis. 

### 3.4. Functional Classification of DEGs

Of the 2495 deregulated DEGs at asymptomatic stage, 1343 were significantly upregulated and 1152 downregulated. On the other hand, in symptomatic stage, of the 4559 deregulated DEGs genes, 1764 were upregulated and 2795 downregulated (Figure 6). Only 936 genes were commonly deregulated between the asymptomatic and symptomatic stages: 369 were upregulated and 530 downregulated. The remaining 37 genes showed inverse behavior between stages: 24 genes were upregulated at asymptomatic stage but downregulated at symptomatic stage, whereas 13 genes were downregulated at asymptomatic stage and upregulated at symptomatic stage (Figure 6). Gene ontology (GO) assignments were used to classify the functions of DEGs at both early and late stages of HLB infection. A total of 25 functional groups at both stages were classified in Biological process, 10 in Molecular function, and 13 in Cellular component. The functional categories with the highest number of genes (>400 genes) were cellular process, metabolic process (from Biological process), binding, catalytic activity, and transferase activity (from Molecular function), and cell, cell part, cytoplasm, intracellular, membrane, organelle, and plastid (from Cellular component). Remarkably, for all categories, a higher number of genes were deregulated at symptomatic stage compared with the deregulated at asymptomatic stage (Appendix A). After that, we performed a GO terms enrichment analysis. At asymptomatic stage, the most significantly enriched GO terms for Biological process were related to response to stimulus (GO:0050896), generation of precursor metabolite and energy (GO:0006091), lipid metabolic process (GO:0006629), and cellular homeostasis (GO:0019725). For GO terms in Molecular function, the most significantly enriched GO terms were catalytic activity (GO:0003824), transferase activity (GO:0016772 and GO:0016740), and hydrolase activity (GO:0016787), and for Cellular component, endomembrane system (GO:0012505) and ribosome (GO:0005840) (Figure 7). On the other hand, at symptomatic stage, the most significantly enriched GO terms for Biological process were response to abiotic stimulus (GO:0009628), generation of precursor metabolites and energy (GO:0006091), embryo development (GO:0009790), and lipid metabolic process (GO:0006629). For GO terms in Molecular function, these were: catalytic activity (GO:0003824), transferase activity (GO:0016740 and GO:0016772), and hydrolase activity (GO:0016787), and for Cellular component, thylakoid (GO:0009579), endomembrane system (GO:0012505), and plastid (GO:0009536) (Figure 7).

### 3.5. Gene Pathway Enrichment Analysis of Host Pathways in Response to CLas Infection

PageMan [30] analysis tool was used to obtain a statistics-based overview of pathways altered during the different stages in response to CLas infection (Figure 8). PageMan can pinpoint upregulated and downregulated expressed genes onto different metabolic and cell function pathways. A total of 10 metabolic pathways were identified through PageMan analysis. Increased expression of genes related to Krebs cycle (TCA and minor carbohydrate metabolism), amino acid (central amino acid, glutamate, proline, and aspartate metabolism), secondary metabolism (isoprenoids, phenylpropanoids, flavonoids, and phenols), transport (v-ATPases, ABC, and multidrug resistance systems and major intrinsic proteins), and hormone metabolism (abscisic acid metabolism and ethylene degradation), was observed at both asymptomatic and symptomatic stages. Transcriptional repression was predominantly observed in photosynthesis (photosystem I and II, LHC complex, electron carrier, light reaction NADH, and cyclic electron flow, and Calvin cycle) and redox (tetrapyrrole synthesis, redox glutaredoxins and nucleotide metabolism) at late stage. For cell wall, a subset of genes were repressed (Arabinogalactan proteins, AGPs) and the other subset induced (lipid metabolism fatty acid desaturation) at early stage, whereas a subset of genes from the same category were only induced at late stage (cellulose synthesis, pectin esterase, and lipid metabolism fatty acid desaturation). Biodegradation of xenobiotics category showed both induction and repression of genes at both stages. At asymptomatic stage, genes related to biodegradation of xenobiotics, cytochrome p450, regulation of transcription zinc finger and WRKY family, and protein synthesis were upregulated, whereas regulation of transcription JUMONI family and protein degradation and folding were downregulated. At symptomatic stage, members of genes related to peroxidases, helicases, regulation of transcription zinc finger and chromatin remodeling factor family, protein posttranslational modification, and ubiquitin-dependent degradation were upregulated, and the genes belonging to rhodanase, regulation of transcription NAC domain and sigma-like family, and protein synthesis, folding, and assembly were downregulated. Finally, a group of genes not assigned showed a repression trend at both stages. After that, MapMan software was applied to display and analyze the functional classes of metabolic pathways significantly different at both early and late stages in response. MapMan provides an overview of pathways and functions identified by PageMan, where each colored square represents a single annotated gene in particular pathway. As shown in Figure 9, at asymptomatic stage, secondary metabolism of isoprenoids, cell wall proteins (arabinogalactan proteins), light reactions, amino acid metabolism, and lipid metabolism are the major categories in which genes are differentially expressed. At symptomatic stage, light reactions, Calvin cycle, cell wall (pectin esterases and cellulose), lipid metabolism (glycolipid synthesis and fatty acid desaturation), nucleotide metabolism, secondary metabolism (phenols and phenylpropanoids), and amino acid metabolism (aspartate) were deregulated. 

#### 3.5.1. Analyzing DEGs Related to Stress Response at Asymptomatic Stage

The MapMan analysis resulted in 164 DEGs related to disease response at the asymptomatic stage (Figure 10). The annotation and expression statistics are summarized in Appendix A. The 164 DEGs were grouped in five categories:
1.Secondary MetabolismA total of 54 DEGs (37 upregulated and 17 downregulated) were observed in this category: 14 genes involved in isoprenoids metabolism, 16 from phenylpropanoids metabolism, 4 miscellaneous alkaloid-like, 6 sulfur-containing proteins involved in glucosinolates and allinases synthesis, 2 involved in wax synthesis, 11 belonging to flavonoids metabolism, and 1 related to phenol metabolism.2.SignalingSixty-five DEGs belonging to signaling category were mostly downregulated at asymptomatic stage. Three genes involved in sugar and nutrient physiology, 37 receptor kinases (20 containing leucine rich repeats I, II, III, VII, VIII, IX, X, XII, and XIV, 1 Catharantus roseus-like RKL1, 6 containing DUF 26 domain, 1 legume-lectic domain, 2 S-locus glycoprotein-like, 4 wall-associated, and 3 miscellaneous kinases, 8 genes related to calcium signaling, 3 phosphoinositide phospholipase C, two G-proteins, 3 MAP kinases, 2 of phosphorelay complex, 5 related to light, and 2 unspecified genes.3.Cell WallTwenty-seven genes (7 up-regulated and 20 down-regulated) were identified in this category. One UDP–glucose epimerase (*UGE*), one UDP–glucuronate synthase (*UXS*), three cellulose synthase genes, one cell-wall-degradation-related protein (beta-mannan endohydrolase), and one gene involved in cell wall modification (*ATEXP1 expansin*) were induced. On the other hand, one *UGE* gene, six cellulose synthases (*CESA4*, *CESA6*, *CESA7*, *CESA8*, *CLB05* and *COBL4*), two genes related to hemicellulose synthesis (*FRA8*, and *IRX9*), three fasciclin-like arabinogalactan proteins (*FLA6*, *FLA11*, and *FLA12*), two cell-wall-degradation-related protein (beta-mannan endohydrolase), three genes involved in cell wall modification (xyloglucosyl transferases, *XTH* genes), and three pectin esterases were downregulated. 4.RNA Regulation of Transcription FactorsFour genes encoding transcription factors containing DOF zinc finger domain (*ADOF2*, *CDF3* and two DOF-type proteins) and four transcriptions containing WRKY domain (*WRKY 23*, *27*, *43*, and *57*) were upregulated.5.Hormone MetabolismTen genes related to abscisic acid metabolism were deregulated. Eight genes were upregulated. Two zeaxanthin epoxidases (*ABA1* genes), two involved in signal transduction and four abscisic-acid-induced genes, were upregulated, whereas, two genes (*NCED4* and *GCR2*, involved in synthesis and signal transduction, respectively) were downregulated.


#### 3.5.2. Analyzing DEGs Related to Stress response at Symptomatic Stage

On the other hand, the 122 genes from the symptomatic (Figure 10 and Appendix A) stage were also grouped in three categories:
1.Secondary MetabolismA total of 71 genes were deregulated (33 upregulated and 48 downregulated) in this category: Nineteen genes involved in isoprenoids metabolism, 16 of phenylpropanoids-dependent lignin biosynthesis, 2 miscellaneous alkaloid-like genes, 7 of glucosinolates synthesis, 3 of wax metabolism, 22 of flavonoids metabolism, and 2 of phenol metabolism.2.RNA Regulation of TranscriptionThree genes encoding transcription factors containing DOF zinc finger domain (including the ADOF2 gene upregulated at early stage and two different DOF-type proteins) were upregulated.3.Redox MetabolismSeven peroxidases were deregulated in this category: five upregulated (*PER22*, *PER32*, *PER34*, and two putative pexosidases genes) and two downregulated (*PER17* and *APX6* genes). A total of 42 redox-related genes were deregulated, most of them repressed (8 upregulated versus 34 downregulated). Fourteen thioredoxin-related genes, 14 glutatione/ascorbate-related, 5 glutaredoxins, 4 peroxiredoxins, 1 superoxide dismutase, 2 catalases, 1 heme group-containing protein, and 1 sulfiredoxin.


Understanding the expression differences of defense-related genes at both early and late stage of the disease may help to elucidate the host response before the disease is well-established. 

## 4. Discussion

Huanglongbing (HLB), or greening disease, vectored by ACP (Asian citrus psyllids), represents a major threat for citrus production worldwide. Efforts to obtain tolerant citrus cultivars through conventional breeding have been impaired by the lack of resistant genes in known citrus species [4]. To overcome this limitation, transgenesis and genome editing through CRISPR technology represent promising alternatives [10]. However, application of these technologies requires basic knowledge of molecular host–pathogen interaction. Transcriptomic profiling is the first step to identify target genes with potential to be applied in the design of opportune diagnostic methods and new management integral strategies. Despite some citrus species, including Mexican lime, being described as tolerant to HLB [11,12,13], studies focused on identifying genes involved in tolerance are limited [17,19,20,21]. 

In the present study, we have obtained a transcriptomic profile of CLas-infected Mexican lime plants in response to HLB disease, at early (asymptomatic) and late (symptomatic) stages of HLB disease development. Previous transcriptomic studies of HLB-citrus response/interaction have been performed mainly at a symptomatic stage where important pathogen-guided host modulation, as well as host defensive responses, could be masked by pleiotropic effects of late infections. An important aspect of disease progression during plant–microbe interactions is the host responses induced at early (asymptomatic) stages. Identification of host responses, especially at asymptomatic stages, can be critical for understanding the initial steps of the disease, and this knowledge could be exploited to design novel diagnostic methods as well as efficient management practices [32]. For RNA-Seq analysis, leaves from CLas-infected plants (*n* = 5) containing homogeneous titer of bacteria (~2.5 × 10^2^ bacterial cells/100 ng of total DNA at asymptomatic stage and with ~1.4 × 10^4^ bacterial cells/100 ng of total DNA at symptomatic stage) and disease symptoms were used for library construction. Previous transcriptomic studies of citrus-HLB interaction were performed using a low sample number (*n* = 3), and in some cases, HLB-infected plants were diagnosticated by end-point PCR or based on a range of Ct detection from qPCR quantification, introducing low biological representation and therefore biasing analysis [18,19,21,33]. In our study, we used two biological replicates (*n* = 5 each replicate) for library construction, ensuring homogeneity in both symptom development and bacteria titer and therefore enhanced reliability of the transcriptomic results. Additionally, even though we have used a pooling strategy for library construction, validation of 10 selected genes by RT-qPCR has been established for individual plants from each library, with transcription levels obtained for individual plants being fully comparable with the data obtained from RNA-Seq analysis in the pooled plants (Figure 5). A total of eight libraries were obtained and deep sequenced by Illumina platform obtaining 21–29 million 300 bp paired-end reliable reads per library. 

Previous transcriptomic profiling studies in response to HLB using microarray technology were performed with an Affymetrix Citrus Genome Array, which is based on 33,879 EST sequences from some citrus species and hybrids. On the other hand, high-throughput sequencing (HTS) technology has been applied using as reference the sweet citrus available genomes of *C. sinensis* and *C. clementina* [9,19,33]. In both cases, the reference probes or genomes can introduce biased data for reads mapping, resulting from unspecific hybridizations, and could cause the loss of species-specific sequences. Therefore, in order to obtain an accurate mapping of generated reads, we used as reference our recently reported species-specific transcriptome of *Citrus aurantifolia* [26], and the corresponding orthologs were annotated from *A. thaliana* TAIR10 database. Then, we carried out a differential expression analysis by means DEgenes-Hunter [28]. A principal component (PC) analysis showed a correlation between biological replicates demonstrating the reliability and consistency of transcriptional data (Figure 4). As a result of differential expression analysis, a total of 2495 CLas-dependent differentially expressed genes (DEGs) were deregulated during asymptomatic stage (1343 upregulated and 1152 downregulated), whereas, in symptomatic stage, 4559 genes were significantly affected (1764 were upregulated and 2795 downregulated) (Figure 3c). Our results are in general agreement with previously described transcriptomic analysis in response to HLB infection, where the range of DEGs was 56–1953, and 22–1523 for up- and downregulated genes, respectively [9,14,15,16,17,18,19,21,33,34,35]. Gene ontology (GO) assignments were used to classify the functions of DEGs at both early and late stages of HLB infection. GO terms enrichment analysis showed that for the Biological process category, the GO terms corresponding to response to stimulus, generation of precursor metabolite and energy, and lipid metabolic stress are the most enriched terms for both asymptomatic and symptomatic stages. Similarly, for Molecular function category, catalytic activity and transferase activity, and for Cellular component category, endomembrane system, were the terms more enriched, in both stages (Figure 7). 

It is known that some HLB-caused symptoms are related to disturbance of both carbohydrate and photosynthesis processes. HLB causes swelling of middle lamella between cell walls surrounding sieve elements in non-symptomatic new flushes and the necrosis of sieve element and companion cells of phloem by callose plugging and excessive starch and, consequently, a disruption of starch metabolism and the inhibition of transport of photosynthates [33,36]. Previous transcriptomic studies of citrus response to CLas infection have reported that carbohydrate metabolism was significantly affected and photosynthesis-related genes were downregulated [9,17,33,37]. In the present study, a significant deregulation of carbohydrate metabolism genes was not observed. Only a few genes of minor metabolism of carbohydrates were upregulated at asymptomatic stage. This behavior could be due to sampling time (8 and 16 wpi) being too soon to observe the carbohydrate metabolism disruption. In agreement with this assumption, we previously observed a significant increment in starch content in leaves of CLas-infected Mexican lime at 48 wpi [38]. The delayed response related to carbohydrate metabolism shown by Mexican lime could be part of a tolerance mechanism, avoiding the early phloem blockage caused by callose and starch deposition. On the other hand, photosynthesis-related genes were downregulated as expected according to previous reports. Members of photosystem I and II, LHC complex, electron carrier, light reaction NADH, cyclic electron flow, and Calvin cycle were downregulated. 

The plant cell wall provides structural support and constitutes the first barrier against microbial invasion. It is known that host susceptibility to pathogens depends on the cell wall composition and structure, which determines its resistance to degradation by cell-wall-modifying enzymes encoded by invading pathogens [39]. In our study, as response to CLas infection, a total of 27 genes related to cell wall metabolism were deregulated, most of them downregulated at asymptomatic stage. The downregulation of a subset of cellulase synthases (*CESA4*, *CESA6*, *CESA7*, *CESA8*, *CLB05*, and *COBL4*) is in agreement with previous reports [20,21], where was suggested that the downregulation of these enzymes could be related with cell wall strengthening and consequently with the tolerance displayed by Kaffir lime (*C. hystrix*) and “Jackson” grapefruit. We also observed the downregulation of three pectin esterases. The downregulation of these enzymes has been observed in both tolerant species “Jackson” grapefruit, and Rough lemon (*C. jambhiri*) [17,20]. On the other hand, three fasciclin-like arabinogalactan proteins (AGPs) (*FLA6*, *FLA11*, and *FLA12*) were also downregulated. AGPs play defensive roles against pathogenic infection by secretion and clumping off AGPs at the affected sites, limiting pathogen propagation [40]. Therefore, the downregulation of these genes could be related to CLas-mediated defensive evasion at early stages of the HLB disease in Mexican lime. 

Secondary metabolites play essential roles in plant defense. Previous studies have shown that most genes involved in secondary metabolism, including terpenoid, phenylpropanoid, and flavonoid biosynthesis pathways, are mostly induced in CLas-infected leaves [18,19,20]. Terpenoids are a structurally diverse group of natural products, providing to plants antioxidant and insect-attracting/-repellent activities [41]. Phenylpropanoids are structural polymers providing protection from pest and UV light and act as pigment pollinator attractants [42]. Flavonoids possess antifungal and antioxidant activities [43]. In the present study, most of the secondary-metabolism-related genes were upregulated (37/54) at asymptomatic stage, possibly as part of early host response against infection. Recent studies have shown that tolerant citrus cultivars contain relatively higher specific volatile compounds, including terpenes and aldehydes, compared with susceptible cultivars [44]. Additionally, it has been reported that tolerant Kaffir lime leaves contain different classes of terpenoids and phenylpropanoids, which are known to possess antimicrobial activities [45,46]. It would be interesting to determine the metabolomic profile of Mexican lime leaves during CLas infection, in order to identify compounds associated with the response and tolerance against this pathogen.

Reports have shown hormone pathways to be strongly related to plant disease progress, blocking or promoting the pathogen proliferation. In both susceptible and tolerant cultivars, phytohormone metabolism is deregulated. In CLas-infected susceptible cultivars, different phytohormone pathways, including ethylene, cytokinin, auxin, gibberellins, and abscisic-acid-related genes were down-regulated, whereas jasmonic-acid-related genes were upregulated [18,33]. In the present study, deregulation of abscisic acid (ABA)-related genes was observed at asymptomatic (early) stage of infection, whereas no significant DEGs related to hormone metabolism was observed at symptomatic (late) stage. A total of eight genes involved in abscisic acid synthesis, sensing, and signaling were upregulated. ABA can promote resistance against bacteria through its ability to induce stomata closure, blocking pathogen entry. On the other hand, ABA may promote pathogen virulence, since increasing ABA signaling or exogenous hormone application of ABA induces *Pseudomonas syringe* proliferation [47]. According to this observation, upregulation of ABA-related genes at early stage in CLas-infected Mexican lime plants could be related to pathogen modulation of host response in order to initiate plant colonization. In CLas-infected tolerant Rough lemon, genes related to ABA metabolism were upregulated, whereas genes involved in the ethylene pathway were upregulated in susceptible infected sweet orange [17]. These results suggest that tolerant citrus species share some hormone-related pathways in response to CLas infection. In contrast, in symptomatic Mexican lime infected with phytoplasmas, a significant upregulation of genes related to the biosynthesis of gibberellins (GA) has been observed. This deregulation of genes related to GA biosynthesis and signaling pathways suggests that GA might be a key hormone that contributes to the development of witches’ broom symptoms. In addition, in the same study, an upregulation of genes related to cell-wall biogenesis and degradation, innate immunity, and secondary metabolism was observed [48]. 

At asymptomatic stage, among the 65 DEGs belonging to signaling category, 37 were receptor-like kinases (RLKs). RKLs are transmembrane receptors similar to animal tyrosine kinases receptors and play important roles in plant disease resistance [49]. Interestingly, 34 of 37 deregulated RKLs were downregulated. This data suggests a bacterial anti-defensive mechanism in order to evade early host responses. 

At asymptomatic stage, four transcription factors containing WRKY domain were induced. Transcription factors (TFs) are regulatory proteins that have the ability to alter gene expression reprogramming, plant growth, and plant defense mechanisms. Among the TFs, WRKY is the most responsive in plant defense [50]. WRKY upregulation was observed in CLas-infected susceptible *C. sinensis* and red tangerine; however, no significant deregulation was observed in tolerant rough lemon [17,33]. These results suggest that WRKY deregulation is not a differential response between susceptible and tolerant citrus cultivars. On the other hand, five transcription factors belonging to DOF zinc finger domain were upregulated at both asymptomatic and symptomatic stages. TFs belonging to DOF superfamily contain an activation domain located at finger structure, and a nuclear localization signal. These TFs are involved in the establishment and adaptation of vascular system [51]. For the DOF-related TFs identified in the present study, only for *CDF3* has vascular localization been confirmed [52]. It would be interesting to determine whether the other DOF-related TFs identified in the present study are located at vascular tissues and to determine their putative role in Mexican lime vascular architecture adaptation in response to CLas infection. 

In plants, after pathogen attack, the passive defense mechanism mediated by structural barriers or existing antimicrobial compounds prevent host colonization, while the active or induced defense responses include hypersensitive response (HR) and systemic acquired resistance (SAR). These processes are guided by signaling molecules, including phytoalexins, pathogenesis-related (PR) proteins, reactive oxygen species (ROS), and reactive nitrogen species (RNS). Consequently, lignification and reinforcement of cell wall occur. Among the proteins induced during plant defense, the plant peroxidases play roles in reinforcement of cell wall, enhanced production of ROS, and increased production of phytoalexins [53]. We observed the upregulation of five peroxidases at symptomatic stage of Mexican lime CLAs-infected leaves, suggesting they have a role as part of induced defensive mechanism against CLas infection. Induction of peroxidases has been reported previously in response to CLas infection in susceptible *C. sinensis* [9]. Large inductions of ROS production after extreme environmental stresses (including pathogen attack) result in oxidative damage and ultimately trigger the cell death. Plant cells possess detoxifying systems to scavenge ROS followed by several antioxidant enzymes, including superoxide dismutase (*SOD*), catalase (*CAT*), glutathione peroxidase (*GPX*), and peroxiredoxin (*Prx*) [54]. SOD was upregulated in tolerant kaffir lime, but downregulated in susceptible *C. sinensis* [21]. At the late stage of CLas infection in Mexican lime, we observed two *SOD*, one *CAT*, and four *Prx* genes, all of them repressed. The downregulation of these genes could be associated with ROS species accumulation, leading to cell death and expression of symptoms associated with HLB. Elucidating whether ROS scavenger-related proteins repression is a pathogenic strategy or a pleiotropic effect of the host-CLas interaction could help in the understanding of the disease. 

## Figures and Tables

**Figure 1 microorganisms-08-00528-f001:**
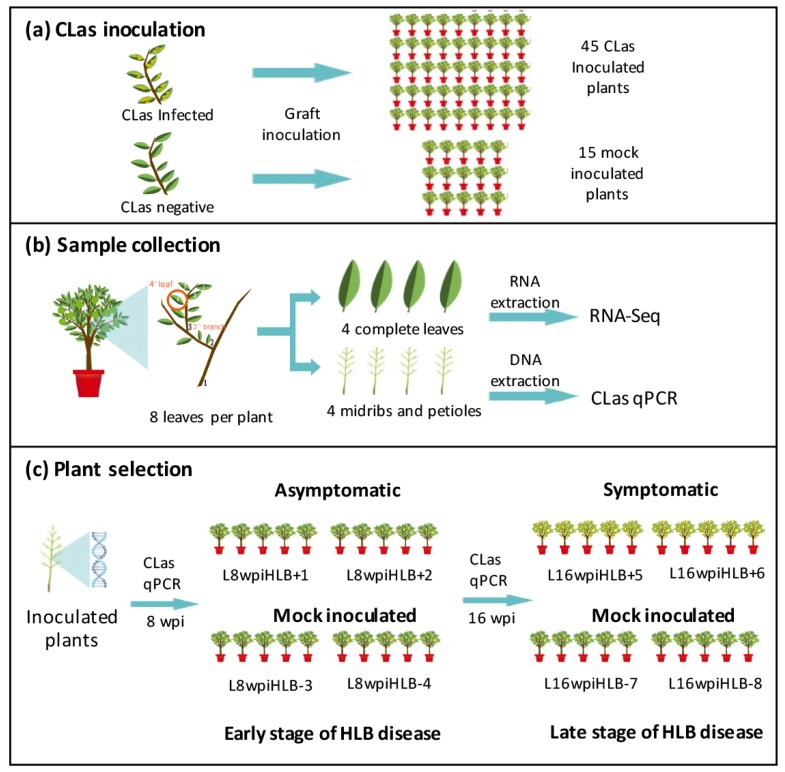
Flowchart of the experimental design used to inoculate *Candidatus* Liberibacter asiaticus (CLas) in *C. aurantifolia* and to select plants to be included in the 8 libraries for RNA-seq analysis. (**a**) CLas inoculation by grafting. Forty-five Mexican lime plants were inoculated with CLas-positive budwood, and 15 plants were inoculated with CLas-free budwood as controls; (**b**) sample collection of 8 leaves per plant for RNA and DNA extraction. In all cases, the fourth leaf of 8 different branches located at third level of ramification from the main stem was collected in order to obtain a homogeneous representation of source tissue; (**c**) plant selection by qPCR at 8 and 16 wpi (weeks post inoculation) to construct each library. Libraries L8wpiHLB+1 and L8wpiHLB+2 included 5 pooled asymptomatic plants. Libraries L16wpiHLB+5 and L16wpiHLB+6 included 5 pooled symptomatic plants. Control libraries, L8wpiHLB-3 and L8wpiHLB-4 at 8 wpi and L16wpiHLB-7 and L16wpiHLB-8 at 16 wpi, were constructed by pooling 5 mock-inoculated plants.

**Figure 2 microorganisms-08-00528-f002:**
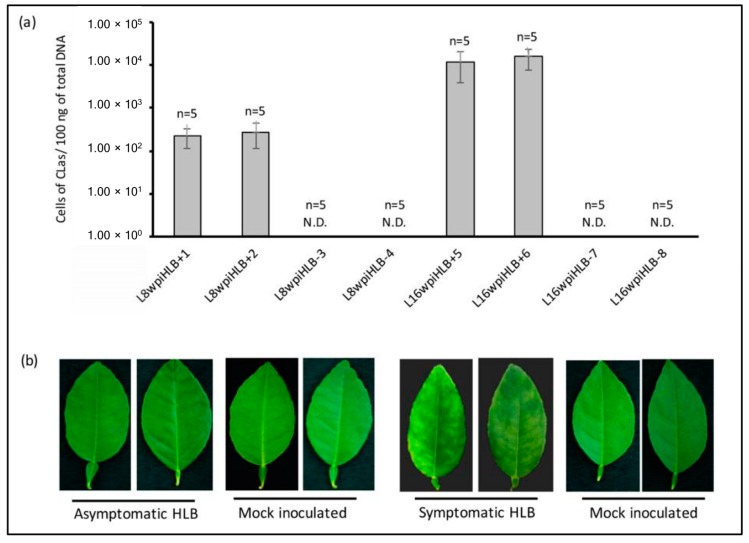
CLas titer quantification and symptoms of selected samples for RNA-seq analysis. (**a**) Bacterial titer in Mexican lime leaves of the five individual plants included in each library at 8 and 16 weeks post-inoculation (wpi). L8wpiHLB+1 and L8wpiHLB+2 are two replicates of asymptomatic plants at 8 wpi. L16wpiHLB+5 and L16wpiHLB+6 are two replicates of symptomatic plants at 16 wpi. L8wpiHLB-3 and L8wpiHLB-4 are control plants (mock-inoculated) at 8 wpi, and L16wpiHLB-7 and L16wpiHLB-8 are control plants (mock-inoculated) at 16 wpi. For each bar, standard deviation is shown. N.D: not detected. (**b**) Representative pictures of Mexican lime leaves used as source of biological material in the present study.

**Figure 3 microorganisms-08-00528-f003:**
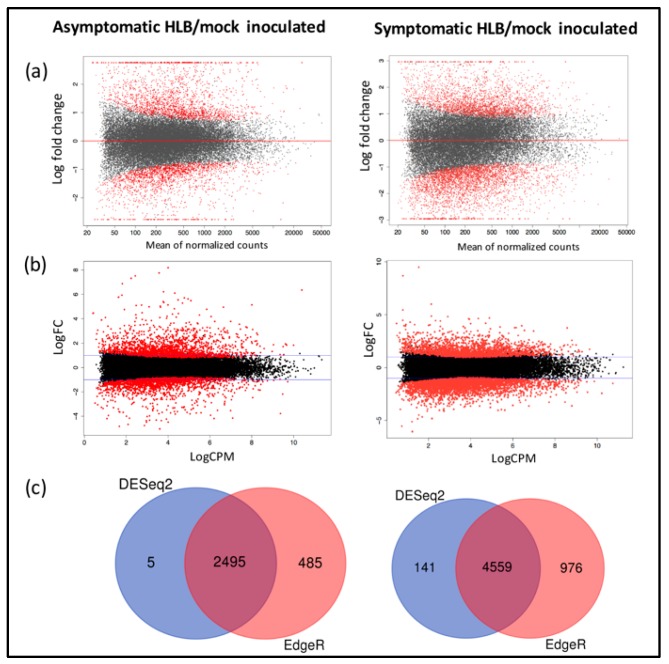
Mexican lime differential gene expression in asymptomatic (early) and symptomatic (late) stages of Huanglongbing (HLB) disease. Log ratio versus abundance plots (MA-plot) for both infection stages using DESeq2 (**a**) and EdgeR (**b**) algorithm. (**c**) Venn diagram of differentially expressed transcripts using both algorithms.

**Figure 4 microorganisms-08-00528-f004:**
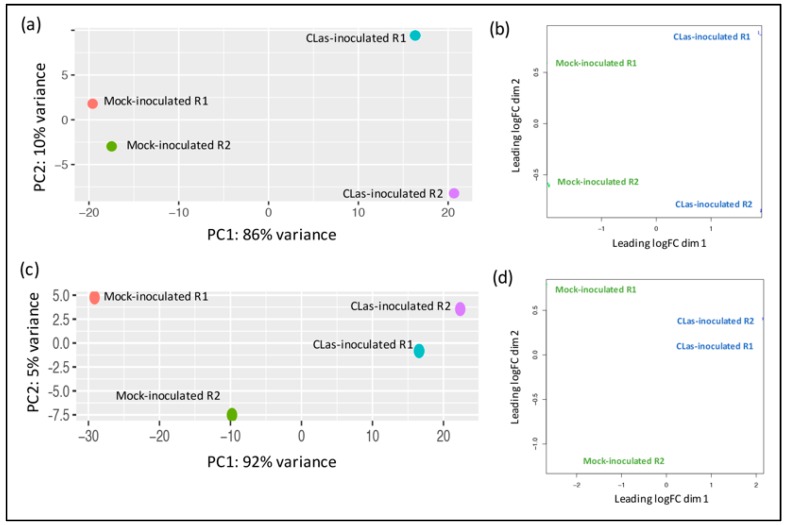
Principal component analysis (PCA) of gene expression from RNA-seq derived data. (**a**,**c**) PCA performed using DESeq2 log-normalized RNA-seq data in asymptomatic and symptomatic stages, respectively. Loadings for principal components 1 (PC1) and PC2 are reported in graph (on x- and y-axes); (**b**,**d**) PCA performed using EdgeR log-normalized RNA-seq data in asymptomatic and symptomatic stages respectively.

**Figure 5 microorganisms-08-00528-f005:**
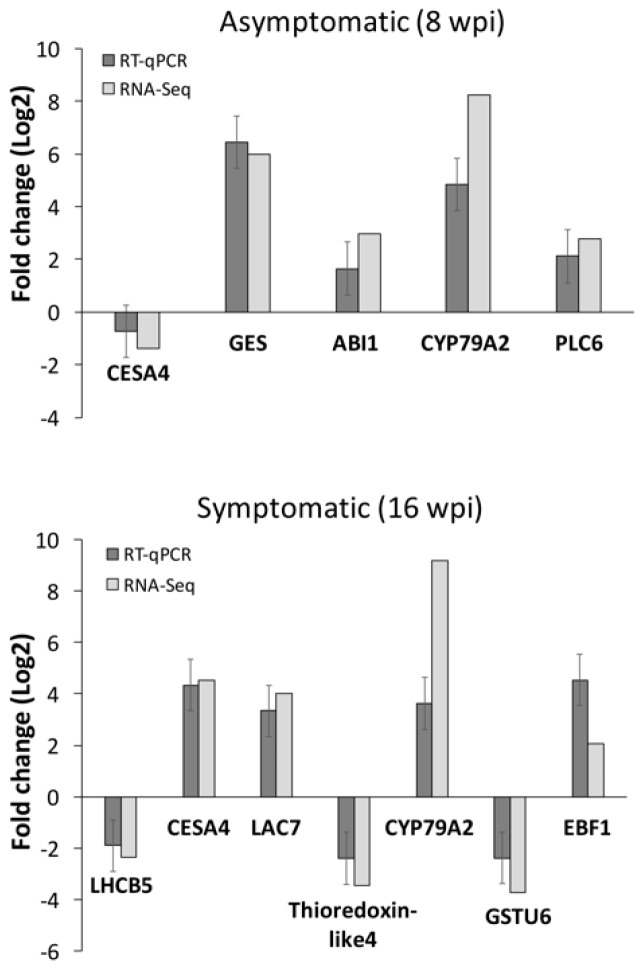
Validation of differential expressed genes. Ten differential expressed genes (DEGs) were selected from RNA-seq data and quantified by RT-qPCR at asymptomatic and symptomatic stages. A comparison between RT-qPCR results and results obtained from RNA-Seq analysis is shown. *CESA4*: Cellulose synthase A catalytic subunit 4; *GES*: (E,E)-geranyllinalool synthase; *ABI1*: Protein phosphatase 2C 56; *CYP79A2*: Cytochrome p450 79a2; *PLC6*: Phosphoinositide phospholipase C 6; *LHCB5*: Chlorophyll a-b binding protein; *Thioredoxin-like 4*; *GSTU6*: Glutathione S-transferase TAU 6; *EBF1*: EIN3-binding F-box protein 1. For RT-qPCR quantification, each bar represents the average of 10 plants. Standard deviation is shown.

**Figure 6 microorganisms-08-00528-f006:**
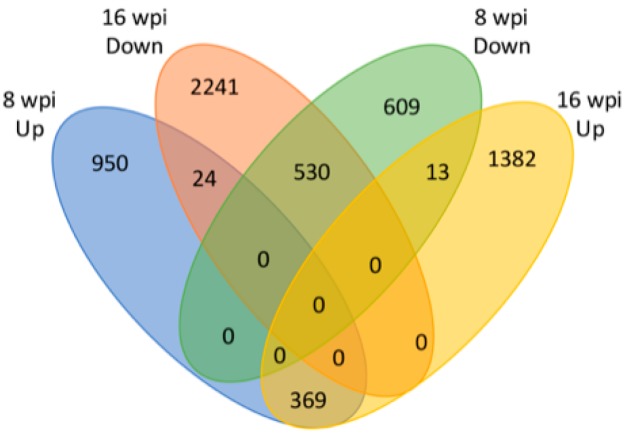
Venn diagram of total number of upregulated and downregulated genes in asymptomatic (8 wpi) or symptomatic (16 wpi) stages of HLB disease.

**Figure 7 microorganisms-08-00528-f007:**
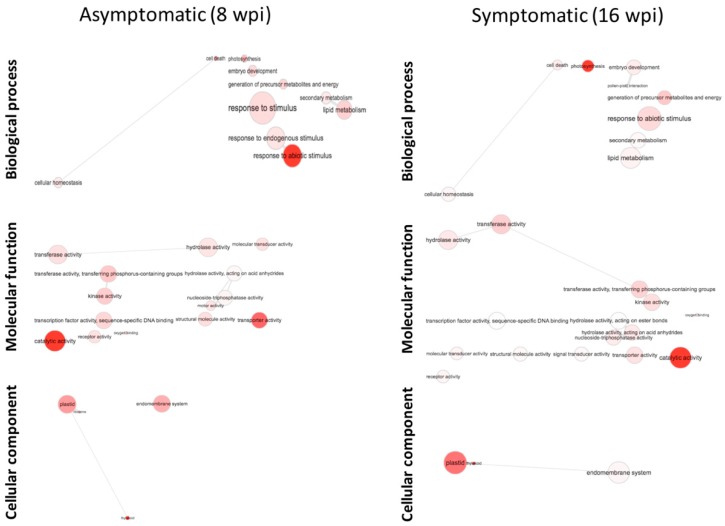
Functional gene-network analysis of DEGs identified by GO enrichment. Interactive graph was generated using web-tool REVIGO (available online: http://revigo.irb.hr/). The web-tool agriGO (available online: http://bioinfo.cau.edu.cn/agriGO/analysis.php) was used to obtain a Singular Enrichment Analysis of the Gene Ontology (GO) terms. In the plot, bubble color and size indicates the *p*-value generated and GO term frequency, respectively. GO terms sharing high similarity are linked by edges, where similarity degree is indicated by line width.

**Figure 8 microorganisms-08-00528-f008:**
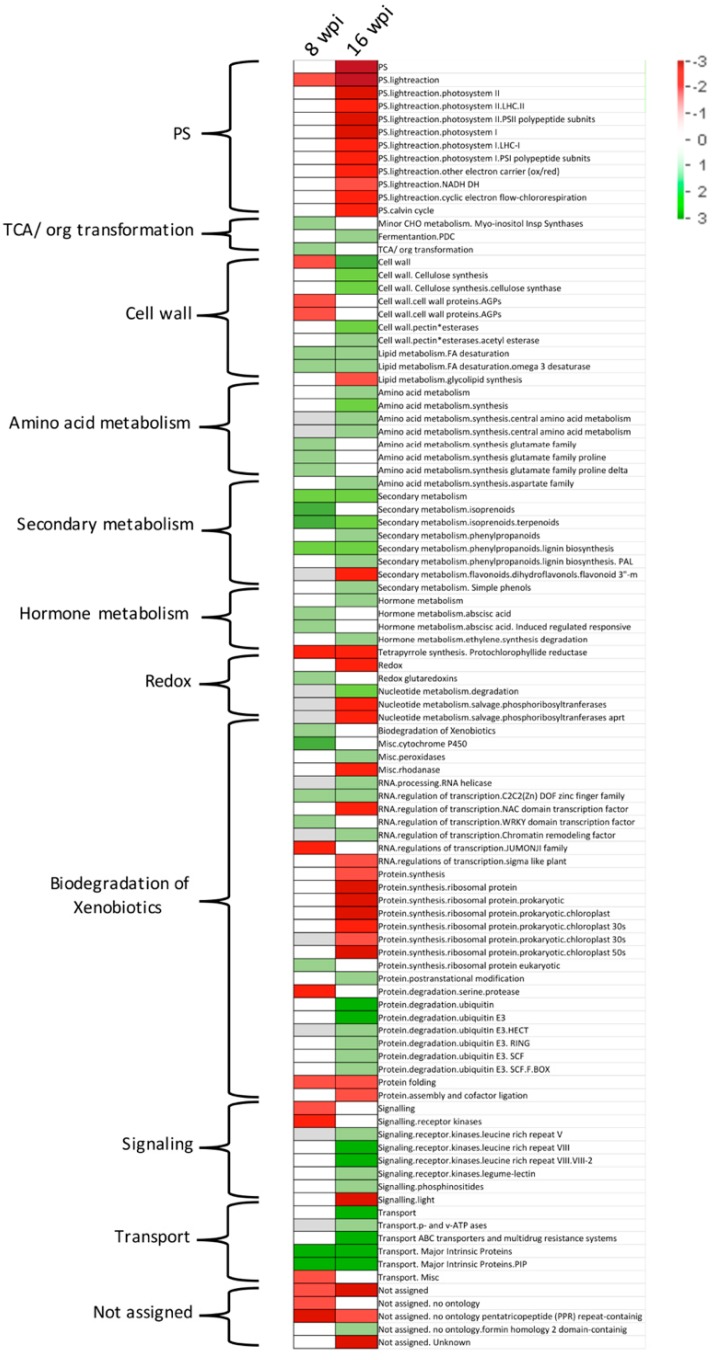
Metabolic pathways categorization analysis. PageMan functional enrichment of differentially expressed genes. Red, downregulated functional categories; green, upregulated functional categories.

**Figure 9 microorganisms-08-00528-f009:**
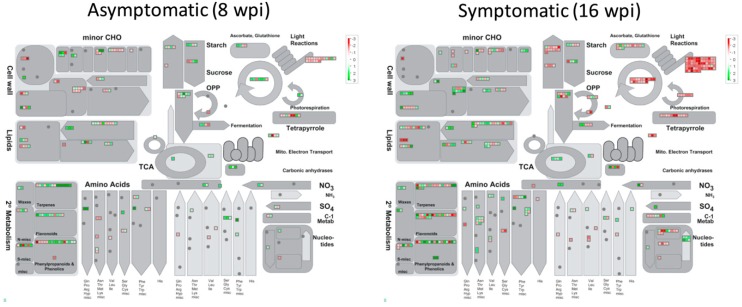
Overview of metabolic pathways that are regulated by CLas infection in Mexican lime (*Citrus aurantifolia*). Genes that were significantly upregulated following CLas infection are displayed in green, and downregulated genes are displayed in red.

**Figure 10 microorganisms-08-00528-f010:**
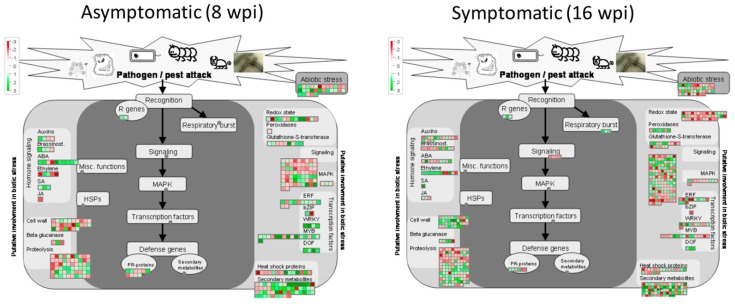
Regulation of stress-related gene pathways by CLas infection in Mexican lime (*Citrus aurantifolia*). Genes that were significantly upregulated following CLas infection are displayed in green, and downregulated genes are displayed in red.

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
