# Peer review of "Gene Expression Profile of Mexican Lime (Citrus aurantifolia) Trees in Response to Huanglongbing Disease caused by Candidatus Liberibacter asiaticus"

_microorganisms, 2020, doi:10.3390/microorganisms8040528_

Round 1

Reviewer 1 Report

The manuscript by Arce-Leal et al investigates the molecular changes in Mexican Lime Tree in response to CLas. Overall, the manuscript is well drafted an the results are discussed extensively. I have some minor comments that might help improving the clarity and impact of manuscript

For clarity, I suggest you include the scientific name of Mexican Lime Tree in the title. And also mention citrus greening disease. Maybe something like
Gene Expression Profile of Mexican Lime Trees Citrus aurantifolia in Response to citrus greening caused by Candidatus Liberibacter Asiaticus

Results section 3.2: A phylogeny should be done on all samples after quality check and adapter trimming (before DEG identification). This will indicate the consistency among biological replicates. The dendrogram can be included in the supplementary material.

Line 237: Figure 2 captions, it is written "... L16wpiHLB-7 and L16wpiHLB-7 are control plants ....". Do you mean L16wpiHLB-7 and L16wpiHLB-8? (check carefully for such typos)

Line 251: "The mapped transcripts, the corresponding A. thaliana orthologs and their annotations are summarized in Supplementary material 2 (Table S2)". This is confusing. Table S2 only provides info on DEGs, not all the mapped transcripts.

Results DEGs: While detecting DEGs, did you detect the expression of housekeeping genes? For example COX gene that was used in RT PCR as well? It's a control that is normally included in RNA Seq studies.

Figure 5: How the values on y axis (relative expression) were calculated for both RNA-Seq and RT-qPCR? This should be explained in the methods

Author Response

Q1: For clarity, I suggest you include the scientific name of Mexican Lime Tree in the title. And also mention citrus greening disease. Maybe something like Gene Expression Profile of Mexican Lime Trees Citrus aurantifolia in Response to citrus greening caused by Candidatus Liberibacter Asiaticus

R1: The titles was changed to: Gene Expression Profile of Mexican Lime (Citrus aurantifolia) Trees in Response to Huanglongbing disease caused by Candidatus Liberibacter asiaticus.

Q2: Results section 3.2: A phylogeny should be done on all samples after quality

check and adapter trimming (before DEG identification). This will indicate the consistency among biological replicates. The dendrogram can be included in the

supplementary material.

R2: The Principal component analysis (PCA) in Figure 4 shows the clustering of the total number of genes from independent conditions (CLas-infected and Mock inoculated plants at both asymptomatic and symptomatic stages) by two packages (DESeq2, and EdgeR). This result confirm the consistency among biological replicates in the different conditions.

However,  the title Figure 4 contains an error since not only DEGs are analyzed but all genes by conditions were included in the analysis.

The caption of Figure 4 was changed to: Line 282: Principal component analysis (PCA) of gene expression from RNA-seq derived data.

Q3: Line 237: Figure 2 captions, it is written "... L16wpiHLB-7 and L16wpiHLB-7 are

control plants ....". Do you mean L16wpiHLB-7 and L16wpiHLB-8? (check carefully for such typos).

R3: The error was corrected.

Line 237: L8wpiHLB-3 and L8wpiHLB-4 are control plants (mock-inoculated) at 8 wpi, and L16wpiHLB-7 and L16wpiHLB-8 are control plants (mock-inoculated) at 16 wpi

Q4: Line 251: "The mapped transcripts, the corresponding A. thaliana orthologs and their annotations are summarized in Supplementary material 2 (Table S2)". This is confusing. Table S2 only provides info on DEGs, not all the mapped transcripts.

R4: We agree that Table S2 only provides info on DEGs. We decided include only DEGs since the complete trascriptome annotation was recently reported by our group (Arce-leal et al. 2020). The sentence was deleted. The description of Table S2 are in lines 260-261.

Q5: Results DEGs: While detecting DEGs, did you detect the expression of housekeeping genes? For example COX gene that was used in RT PCR as well? It's a control that is normally included in RNA Seq studies.

R5: We analyzed the expression level of COX, and other commonly used housekeeping genes including b-Tubulin, and Elongation Factor 1-a (EF1-a). For all genes no significant differences according EdgeR and DESeq2 package (|log2ratio| < 1, and FDR > 0.05, respectively) were observed at both asymptomatic and symptomatic stages, corroborating the stability of housekeeping genes in our experimental conditions.

Q6: Figure 5: How the values on y axis (relative expression) were calculated for both RNA-Seq and RT-qPCR? This should be explained in the methods

R6: We agree that description of RNA-Seq versus RT-qPCR data comparison could be confusing. In fact, we noticed that relative expression term used in Figure 5, do not correspond to the analysis we performed. Instead, fold change term is the appropriate.

For RNA-Seq data, differentially expressed genes (DEGs) are identified after statistical analysis of fold change expression values for each individual gene. Fold change values are obtained from the ratio of expression data between HLB-infected versus the corresponding mock-inoculated control. The obtained fold change data are expressed in Log2 base, and only the genes with values |log2ratio| ≥ 1 are considered DEGs.

In order to compare RNA-Seq with RT-qPCR data, relative expression of selected genes was obtained by the 2-ΔΔCt method. Similarly to RNA-Seq analysis, fold change data resulted from the ratio of RT-qPCR expression data between HLB-infected versus the corresponding mock-inoculated control, and expressed in Log2 base.

We clarified this procedure in Gene Expression Validation section

Lines 210-216 were deleted and this paragraph was included: Relative expression levels of selected target genes were calculated by means the 2-ΔΔCt method [31] using the Cq value of COX gene as internal reference control. For each sample the fold change value was obtained from the ratio of relative expression data between HLB-infected versus the corresponding mock-inoculated control, and the data converted to Log2 base. All qPCR reactions were analyzed using three technical replicates, and results were plotted as the average of fold change expression data from the two biological replicates analyzed for each condition (10 plants), and compared with the corresponding data in Log2 base from RNA-Seq.

In Figure 5, the “y” axis was changed, “relative expression” was deleted, and  “Fold change (Log2)” was added.

Reviewer 2 Report

The authors present a transcriptomic analysis of symptomatic and asymptomatic citrus hosts when infected with a Gram-negative α-proteobacteria from Candidatus Liberibacter sp. They performed what appears to be a properly designed transcriptomic experiment with HBL- and mock-inoculated citrus where they control that the etiology of the disease is as expected. They do a careful analysis of the transcriptomics results and validate them with qPCR. They find that at the transcriptome level the symptomatic host is significantly more affected than the asymptomatic host. The manuscript is quite long and it is hard for the reader that does not have a background in plant physiology to understand the significance of the analysis of the results.  

Since most of the results relate to the host response, the microorganism physiology is missing which may be problematic considering the scope of the journal. As written, the manuscript reads as it was to be published in a plant physiology journal rather than a microbiology one. For example one of the main conclusions of this comparative analysis is: "The delayed response related to carbohydrate metabolism showed by Mexican lime could be part of tolerance mechanism, avoiding the early phloem blockage caused by callose and starch deposition."     

My suggestions are oriented towards making the manuscript more of interest to the readers of Microorganisms as well as general comments about how to emphasize scientific soundness:

  • In the introduction, the relevance of studying this disease at the molecular level is very well introduced. However, the novelty of the results is not as clear since they mention that studies like this have been done on other species and it is not clear why they expect to find new insights in this experiment. From their discussion, it seems that the authors consider that the previous experiments were less informative because of the number of replicates and the number of transcripts identified. If that is the motivation of this study this should be stated in the introduction. 
  • How many of the responses that are reported here are expected depending on the microorganism physiology? I would suggest the authors incorporate in the discussion some aspects of bacterial physiology, that can potentially relate to the findings on the host physiology that the authors report here. 
  • To introduce the microorganism perspective to the discussion, I would suggest that the authors compare the response of citrus to this pathogen, to the documented response to other pathogens to ask to what extent they have seen responses that are pathogen-specific.   

Author Response

Q1. In the introduction, the relevance of studying this disease at the molecular level is very well introduced. However, the novelty of the results is not as clear since they mention that studies like this have been done on other species and it is not clear why they expect to find new insights in this experiment. From their discussion, it seems that the authors consider that the previous experiments were less informative because of the number of replicates and the number of transcripts identified. If that is the motivation of this study this should be stated in the introduction.

R1. To accomplish this recommendation, at the end of the introduction we considered this paragraph:

“All together, these reports suggest that despite some cellular responses being common between tolerant citrus cultivars, other responses are species-specific, highlighting the importance of transcriptomic studies in response to CLas infection in tolerant citrus species. In order to understand how CLas interacts with Mexican lime at a molecular level, a greenhouse rigorous time-course experiment, avoiding environmental factors and individual differences among plants was designed. In this study we report the transcriptomic profile of leaves from the acid lime C. aurantifolia (Mexican lime) challenged against CLas infection at both asymptomatic and symptomatic stages of HLB disease severity”.

Q2. How many of the responses that are reported here are expected depending on the microorganism physiology? I would suggest the authors incorporate in the discussion some aspects of bacterial physiology, that can potentially relate to the findings on the host physiology that the authors report here. 

R2. We already have some paragraphs about this:

In Lines 571-575: On the other hand, ABA may promote pathogen virulence, since increasing ABA signaling or exogenous hormone application of ABA induce Pseudomonas syringe proliferation [48]. According to this observation, upregulation of ABA related genes at early stage in CLas-infected Mexican lime plants could be related to pathogen modulation of host response in order to initiate plant colonization.

In lines 622-624: Elucidate whether ROS scavenger-related proteins repression is a pathogenic strategy or a pleiotropic effect of the host-CLas interaction could help in the understanding of the disease.

Q3. To introduce the microorganism perspective to the discussion, I would suggest that the authors compare the response of citrus to this pathogen, to the documented response to other pathogens to ask to what extent they have seen responses that are pathogen-specific.   

R3. We add the lines 578-584. “By the other hand, in symptomatic Mexican lime infected with phytoplasmas, a significant up-regulation of genes related to the biosynthesis of gibberellins (GA), has been observed. This deregulation of genes related to GA biosynthesis and signaling pathways suggests that GA might be a key hormone that contributes to the development of witches’ broom symptoms. In addition, in the same study, an upregulation of genes related to cell-wall biogenesis and degradation, innate immunity and secondary metabolism, were observed (Mardi et al 2015).”